# Retrotransposon insertions can initiate colorectal cancer and are associated with poor survival

Tatiana Cajuso[1,2], Päivi Sulo [1,2], Tomas Tanskanen[1,2], Riku Katainen[1,2], Aurora Taira[1,2], Ulrika A. Hänninen [1,2], Johanna Kondelin[1,2], Linda Forsström[1,2], Niko Välimäki[1,2], Mervi Aavikko [1,2], Eevi Kaasinen[1,2], Ari Ristimäki [1,3], Selja Koskensalo [4], Anna Lepistö[4], Laura Renkonen-Sinisalo[4], Toni Seppälä [4], Teijo Kuopio[5,6], Jan Böhm[6], Jukka-Pekka Mecklin[7,8], Outi Kilpivaara[1,2], Esa Pitkänen[1,2], Kimmo Palin [1,2] & Lauri A. Aaltonen[1,2]

Genomic instability pathways in colorectal cancer (CRC) have been extensively studied, but the role of retrotransposition in colorectal carcinogenesis remains poorly understood. Although retrotransposons are usually repressed, they become active in several human cancers, in particular those of the gastrointestinal tract. Here we characterize retrotransposon insertions in 202 colorectal tumor whole genomes and investigate their associations with molecular and clinical characteristics. We find highly variable retrotransposon activity among tumors and identify recurrent insertions in 15 known cancer genes. In approximately 1% of the cases we identify insertions in *APC*, likely to be tumor-initiating events. Insertions are positively associated with the CpG island methylator phenotype and the genomic fraction of allelic imbalance. Clinically, high number of insertions is independently associated with poor disease-specific survival.

[1] Applied Tumor Genomics Research Program, Faculty of Medicine University of Helsinki, Biomedicum Helsinki, PO Box 63 (Haartmaninkatu 8), FI-00014 Helsinki, Finland. [2] Department of Medical and Clinical Genetics, Medicum, University of Helsinki, Biomedicum Helsinki, PO Box 63 (Haartmaninkatu 8), FI-00014 Helsinki, Finland. [3] Department of Pathology, HUSLAB, University of Helsinki and Helsinki University Hospital, (Haartmaninkatu 3), FI-00290 Helsinki, Finland. [4] Department of Gastrointestinal Surgery, Helsinki University Hospital, University of Helsinki, (Haartmaninkatu 4), FI-00290 Helsinki, Finland. [5] Biological and Environmental Science, University of Jyväskylä, PO Box 35 (Seminaarinkatu 15), FI-40014 Jyväskylä, Finland. [6] Department of Pathology, Central Finland Health Care District, (Keskussairaalantie 19), FI-40620 Jyväskylä, Finland. [7] Department of Surgery, Jyväskylä Central Hospital, (Keskussairaalantie 19), FI-40620 Jyväskylä, Finland. [8] Department of Health Sciences, Faculty of Sport and Health Sciences, University of Jyväskylä, PO Box 35 (Seminaarinkatu 15), FI-40014 Jyväskylä, Finland. Correspondence and requests for materials should be addressed to L.A.A. (email: lauri.aaltonen@helsinki.fi)

Retrotransposons are transposable genetic sequences that copy themselves into an RNA intermediate and insert elsewhere in the genome. Almost half of the human genome consists of transposon derived sequences[1], however only a few elements remain retrotransposition competent and account for most retrotranspositions[2,3]. Two types of retrotransposons have been identified in the human genome; autonomous and non-autonomous. Autonomous elements, such as Long Interspersed Nuclear Element-1s (LINE-1s) and Endogenous retroviruses (ERVs), provide the required machinery for retrotransposition. On the contrary, non-autonomous elements, such as Alus and SINE-VNTR-Alu (SVAs), require the LINE-1 machinery to retrotranspose[4–7]. In cancer, ~24% of somatic retrotranspositions involve 3′ transduction, a process characterized by mobilization of 3′ flanking sequence which can serve as a unique sequence revealing the insertion origin[8–11].

LINE-1s are frequently repressed by promoter methylation[12] and genome-wide hypomethylation is reported to lead to their activation during tumorigenesis[13,14], thus leading to high retrotransposon activity and genome instability[14–16]. High retrotransposon activity has been reported in several human cancers, especially in tumors arising from the gastrointestinal tract, such as colorectal cancer (CRC)[10,11,17–20]. Somatic insertion density in tumors is higher in closed chromatin and late replicating regions. Among insertions in genes, insertion density is higher in genes with low expression[10,21]. Furthermore, ongoing retrotransposon activity has been reported in CRC[22]. Insertion count is associated with patient age[18] and LINE-1 hypomethylation is associated with poor survival in CRC[23]. LINE-1 insertions in *APC* have been reported in two CRCs, indicating that these insertions may be early tumorigenic events[24,25]. CRC can develop through two distinct pathways; chromosomal instability (CIN) or microsatellite instability (MSI). Most sporadic CRCs follow the CIN pathway, characterized by a large number of chromosomal alterations. Fifteen percent of CRC cases follow the MSI pathway, characterized by a high number of base substitutions and short insertions and deletions[26]. Seventy-five percent of MSI-positive sporadic CRCs are attributed to the CpG island methylator phenotype (CIMP)[27] which is characterized by gene promoter hypermethylation. Although genomic instability pathways have been studied extensively in CRC, the tumorigenic role of retrotransposition is not fully understood. Retrotransposon insertions have been difficult to detect with previous methodological approaches and very few genome-wide studies have been reported. Here, we characterize somatic retrotransposon insertions in 201 CRCs and one colorectal adenoma utilizing whole genome sequencing (WGS), and investigate the associations between somatic retrotransposon activity and clinical characteristics.

## Results

**Genome-wide detection of somatic retrotransposition in CRC.** To characterize the landscape of somatic retrotransposon insertions in CRC we applied TraFiC[10] and DELLY[28] to WGS data from 202 colorectal tumors and matched normal samples. TraFiC was used to detect insertions without 3′ flanking sequence and DELLY was used to detect LINE-1 transductions that were not identifiable by TraFiC. From the 202 tumors, 12 were MSI and 190 were microsatellite stable (MSS) including three ultra-mutated tumors, harboring somatic *POLE* mutations. After strict somatic filtering, we identified a total of 5072 insertions (Supplementary Data 1). We detected 4726 insertions with TraFiC, and 346 transduction calls with DELLY. Based on visual inspection of the paired-end read data on 100 random insertion calls, 76 calls were evaluated as true somatic insertions, giving a false positive rate of 24% (95% confidence interval [CI], 16–34%) (Supplementary Data 1). Additionally, 14 out of 15 3′ transductions from two samples were validated by long-distance inverse-PCR (LDI-PCR) and Nanopore sequencing in a separate study[22]. The mean number of insertions per tumor was 25 (median, 17; interquartile range, 10–31) with high variability among tumors (Fig. 1a). Mean number of insertions in MSS, MSI and the *POLE* ultra-mutated tumors was 25, 34, and 24 respectively. The majority of insertions (99%, 5024/5072) were LINE-1 retrotranspositions, however we also detected 20 SVA, 13 Alu and 15 ERV insertions (Supplementary Data 1). In concordance with previous studies[10,21], insertion density was higher in closed chromatin (1.78 insertions per Mbp) than in open chromatin (0.96 insertions per Mbp) and in late replicating regions (replication time > 0.8, 3.06 insertions per Mbp) than in early

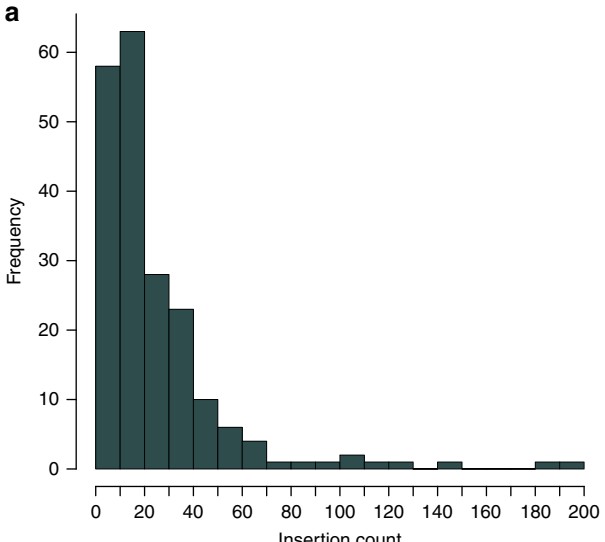

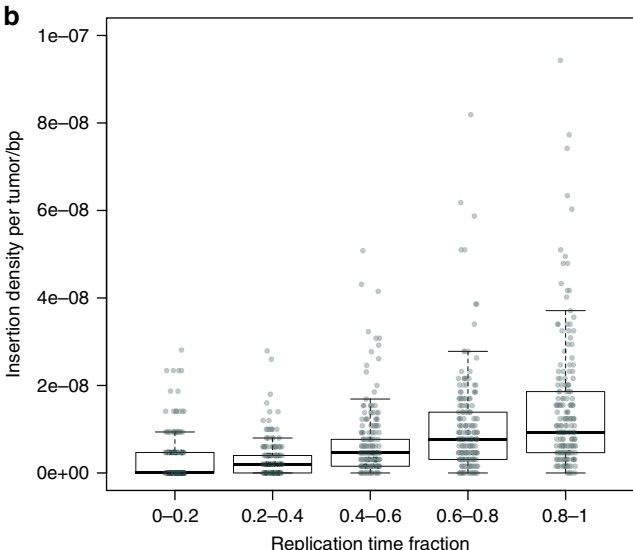

**Fig. 1** Distribution of somatic insertions across 202 colorectal tumors and over replication time. **a** Frequency of somatic insertion counts in 202 colorectal tumors. **b** Insertion density over replication time. The genome was stratified by replication time in five categories where 0 referred to the earliest replication timing. Each point represents insertion density in the corresponding category for each of the 202 tumors. Boxplot shows median, interquartile range (IQR), and whiskers extend to the most extreme data points which are no more than 1.5 times the IQR

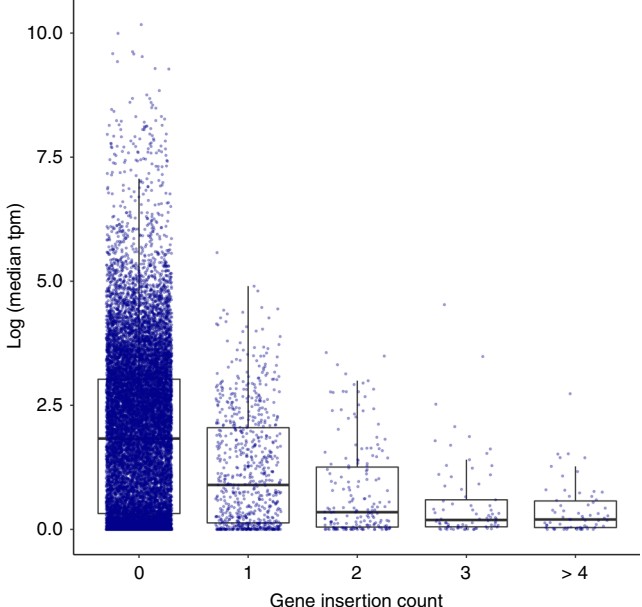

**Fig. 2** Retrotransposon insertions in protein-coding genes. Gene expression (median TPM values from 34 tumors) over gene insertion count groups. Boxplot shows median, interquartile range (IQR), and whiskers extend to the most extreme data points which are no more than 1.5 times the IQR

**Table 1 Genes from the Cancer Gene Census with two or more insertions**

| Gene ID | Gene name | Number of insertions ($n = 202$) | Cancer census role |
|---|---|---|---|
| ENSG00000168702 | LRP1B | 19 | TSG |
| ENSG00000178568 | ERBB4 | 7 | Oncogene, TSG |
| ENSG00000171094 | ALK | 5 | Oncogene, fusion |
| ENSG00000196090 | PTPRT | 3 | TSG |
| ENSG00000046889 | PREX2 | 3 | Oncogene |
| ENSG00000185811 | IKZF1 | 3 | TSG, fusion |
| ENSG00000183454 | GRIN2A | 2 | TSG |
| ENSG00000144218 | AFF3 | 2 | Oncogene, fusion |
| ENSG00000157168 | NRG1 | 2 | TSG, fusion |
| ENSG00000079102 | RUNX1T1 | 2 | Oncogene, TSG, fusion |
| ENSG00000151702 | FLI1 | 2 | Oncogene, fusion |
| ENSG00000134982 | APC | 2 | TSG |
| ENSG00000189283 | FHIT | 2 | TSG, fusion |
| ENSG00000085276 | MECOM | 2 | Oncogene, fusion |
| ENSG00000196159 | FAT4 | 2 | TSG |

Gene names are shown in italics. Cancer census role, role in cancer as defined by the Cancer Gene Census 30
*TSG* tumor suppressor gene

replicating regions (replication time < 0.2, 0.73 insertions per Mbp) (Fig. 1b).

**Retrotranspositions are predicted to initiate ~1% of CRCs.** To characterize retrotransposon insertions in genes, all protein-coding transcripts and the insertion polyA/T in conjunction with gene orientation were used to assess insertion orientation (sense/antisense). Of the 5072 insertions, 1680 (33%) were detected within protein-coding genes, with 98% in introns (Supplementary Data 1, Supplementary Fig. 1). We identified 353 insertions in antisense orientation and 349 in sense orientation (Supplementary Data 1). Insertion count was higher in genes with lower

expression (median transcript per million reads [TPM] from 34 tumors) in concordance with a previous study[21] (Fig. 2).

Recurrent insertions (at least two insertions) were identified in 333 protein-coding genes (Supplementary Data 2) and no significant enrichment of biological processes was observed after correcting for gene length. Fifteen genes in the Cancer Gene Census (CGC)[29] displayed recurrent insertions and no clear bias towards tumor suppressor genes or oncogenes was apparent (Table 1).

The most frequently affected protein-coding genes were *LRP1B* with 19 insertions, *DLG2* with 10 insertions and *PTPRD* and *LSAMP* both with 9 insertions. All the insertions were located in the introns and no insertion clusters were observed (Supplementary Data 1). Higher number of insertions in antisense orientation was observed in *LRP1B* where insertion orientation was available for more insertions (Supplementary Data 1). Both *LRP1B* and *DLG2* have been reported to be fragile sites[30] and recurrent hotspots for HPV integration[31] (Fig. 3). However, no clusters of insertions and HPV integrations nor allelic imbalances (AI) were apparent (Fig. 3).

Genes with highest density of recurrent insertions were *RCN1* with three insertions, and *COL25A1*, *ARAP2,* and *ZNF251* with two insertions. Gene Ontology (GO) annotations associated to these genes include protein binding for *RCN1*; heparin binding and amyloid-beta binding for *COL25A1*; GTPase activator activity, small GTPase binding and phosphatidylinositol-3,4,5-trisphosphate binding for *ARAP2*; and RNA polymerase II transcription activity and DNA binding transcription factor activity for *ZNF251*[32]. None of these genes have been classified as cancer genes[29], fragile sites[30] or hotspots of HPV integration[31]. We also investigated whether insertions had an overall effect on the expression of the closest genes but no significant effect was detected (Supplementary Fig. 2, Methods section Association test between insertions and RNA expression).

Seventy-two insertions were identified in exons of protein-coding genes (Supplementary Data 1, Supplementary Fig. 1). We identified one insertion in the last exon/3′UTR of *PIK3CA* (Supplementary Data 1) and two insertions in exon 16 of *APC* (Fig. 4, Supplementary Data 1). Loss of heterozygosity and copy number loss encompassing *APC* were found in both tumors, and no other sequence variations were identified. Moreover, both insertions were in close proximity (2,151 bp) to two previously reported insertions[24,25]. The location of the insertions was consistent with the distribution of non-synonymous point mutations detected in *APC* and were predicted to disrupt the protein reading frame of *APC* as previously reported[24,25] (Fig. 4). Taken together these findings, as well as the previous extensive knowledge of the tumor-initiating role of *APC* in most CRCs[33,34], suggest that retrotransposon insertions may have contributed to the early steps of tumorigenesis in 2 of the 202 colorectal tumor patients.

**Recurrent insertions in lowly expressed fragile sites**. We observed recurrent insertions in 12 out of 21 genes with high probability of being fragile as estimated in another study[30] (Supplementary Data 3). Since common fragile sites are prone to copy number alterations (CNAs)[35], we evaluated whether retrotransposition and CNAs—in this study detected as AI[36]—were correlated (Supplementary Data 3). Fragile sites with high frequency of insertions seemed to display lower frequency of AI (Fig. 5a). Next, we investigated whether high frequency of insertions (insertion fraction/AI fraction > 1) and high frequency of AIs (0 < insertion fraction/AI fraction < 1) could result from differences in gene expression within fragile sites. Indeed, insertion frequency seemed to be higher in genes with lower

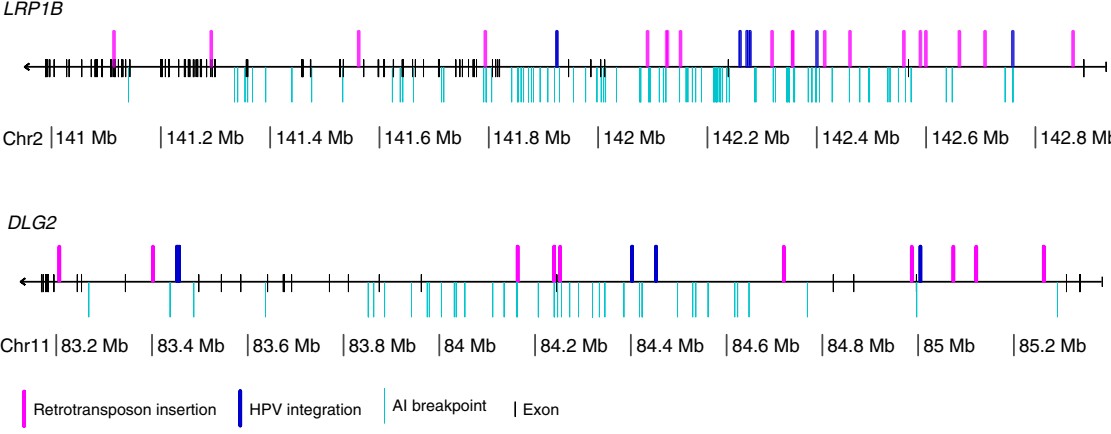

**Fig. 3** Retrotransposon insertion distribution in *LRP1B* and *DLG2*. Mapping of retrotransposon insertions identified in 202 colorectal tumors, HPV integration hotspots reported in 135 cervical cancers and allelic imbalance breakpoints identified in 1,699 CRCs[31,36]. Figure plotted with genoPlotR[64]. Source data are provided as a Source Data file

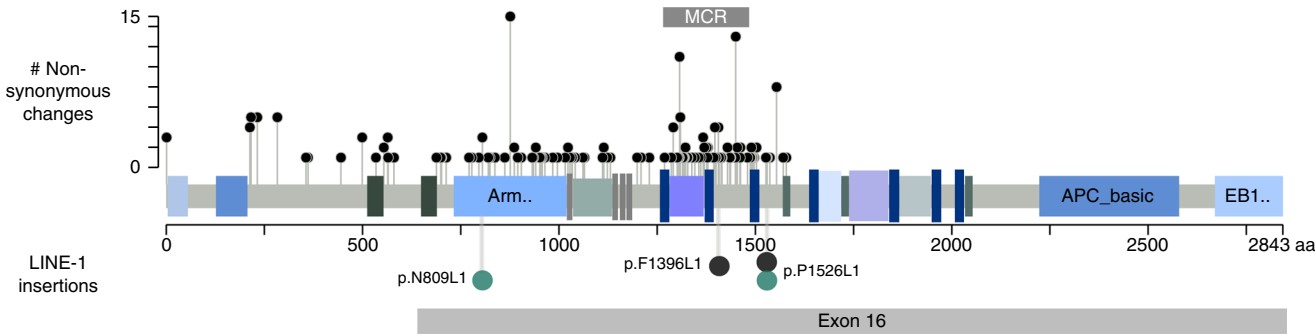

**Fig. 4** Distribution of non-synonymous changes and LINE-1 insertions on the linear protein of *APC*. Non-synonymous changes in 187 MSS CRCs, small lollipops. LINE-1 insertions, larger lollipops. p.N809L1 (c1049.1T) and p.P1526L1 (c310.1T), turquoise lollipops; p.F1396L1 and p.P1526L1[24,25] black lollipops. Figure modified from cBio cancer genomics portal[59,60]

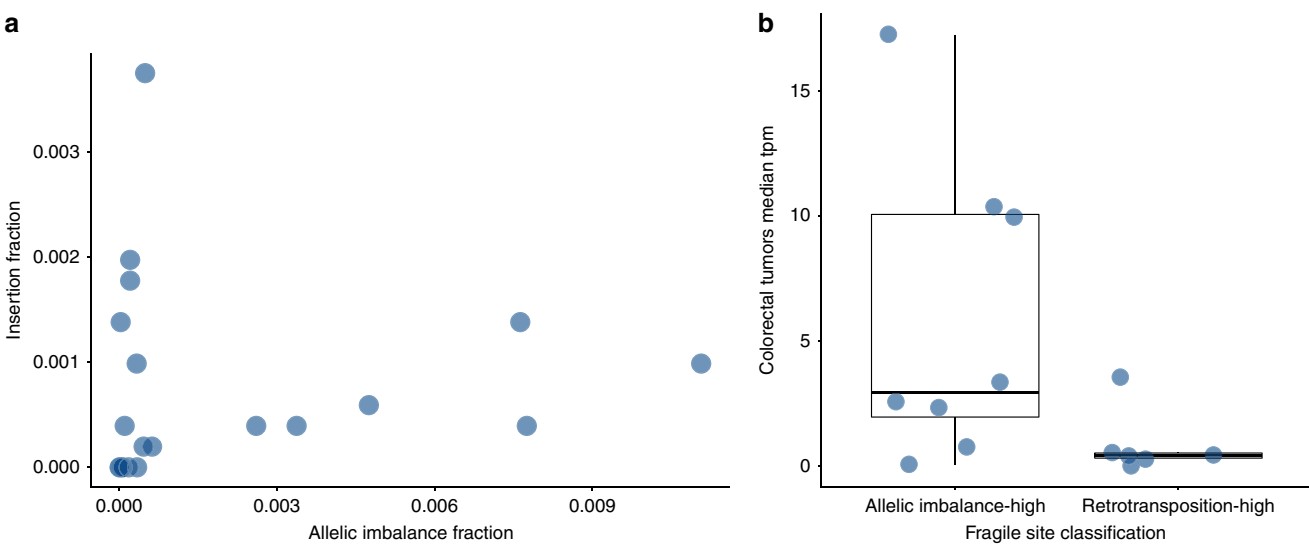

**Fig. 5** Insertion and AI frequency in 21 fragile sites. **a** Insertion fraction over the fraction of allelic imbalance in 21 fragile sites. **b** Gene expression (median TPM values from 34 tumors) in fragile sites with high insertion fraction and fragile sites with high allelic imbalance fraction (Supplementary Data 3). Boxplot shows median, interquartile range (IQR), and whiskers extend to the most extreme data points which are no more than 1.5 times the IQR

expression (Exact Two-Sample Fisher-Pitman Permutation Test for log-transformed gene expression, $p = 0.04$) (Fig. 5b). These results are concordant with our data and those of another study[21]; insertion density is overall negatively correlated with gene expression.

**Few active reference LINE-1s account for most transductions**. We utilized the 3′unique sequence from the transduced regions to identify the reference source elements of LINE-1 transductions. We detected a total of 346 transductions arising from 56 of 315 human specific full-length reference LINE-1s. Fourteen out of the 56 reference elements were previously reported to be active in humans[2,3] and 28 were reported to be active in cancer (Supplementary Data 4)[21]. Recurrent transductions were detected from 24 LINE-1s, and in concordance with our previous study[11] the most active was the LINE-1 located in 22q12.1, which alone accounted for 160 transductions (46%). Seven and six percent of the transductions arose from the LINE-1s located in 9q32 and Xp22.2, respectively. Moreover, the insertion frequencies are in concordance with the frequencies reported by another study across 31 different tumor subtypes (Supplementary Data 4)[21].

**Insertion count associates with CIMP and AI**. We investigated the associations between insertion count and molecular and clinical characteristics. We utilized 196 colorectal tumors with

complete information on molecular and clinical variables that were included in the model (Table 2, Supplementary Data 5). We applied a multiple linear regression model for log-transformed insertion counts, and hypothesized that the number of somatic insertions may be associated with tumor location, *TP53* mutation, MSI, genomic fraction of AI[36] and CIMP. The model was adjusted for mean sequencing coverage, tumor stage, sex and age at diagnosis (Table 2). Goodness-of-fit was tested by Pearson's chi-square test ($p = 0.99$). We found that insertion count was positively associated with CIMP (Multiple linear regression model, $p = 0.00032$) and the genomic fraction of AI (Multiple linear regression model, $p = 0.0036$) (Table 2). Moreover, both associations remained significant after including *BRAF* mutation (V600E) (Multiple linear regression model, CIMP, $p = 0.004$; and genomic fraction of AI, $p = 0.004$) and when only including MSS samples (Multiple linear regression model, CIMP, $p = 0.001$; and the genomic fraction of AI, $p = 0.006$). We also investigated whether insertion breakpoints were located at sites of chromosomal AI (±5000 bp from each AI breakpoint, $n = 40{,}718$) however, only one colocalizing event was identified in one sample (c827, id4279).

**Insertion count associates with poor CRC survival**. We applied the Cox proportional hazards model in 192 patients with complete information on molecular and clinical variables that were used in the model (Table 3, Supplementary Data 5). Patients were followed for 1,370 person-years (Supplementary Data 5). We hypothesized that insertion count may be associated with disease-specific survival (Fig. 6). The model was adjusted for tumor stage, sex, MSI, the genomic fraction of AI, *BRAF* mutation and CIMP status (Table 3). As expected, advanced tumor stage (Dukes C and D) was strongly associated with CRC-specific survival. However, even after adjusting for the above-mentioned covariables, insertion count was independently associated with poor disease-specific survival (Cox proportional hazards model, $p = 0.0029$) (Fig. 6, Table 3).

## Discussion

Although retrotransposon activity is a hallmark of tumors of the gastrointestinal tract[10,11,17–20], the role of retrotransposon insertions in CRC remains unclear with very few studies reported. Here, we characterized the somatic landscape of retrotransposon insertions in the largest dataset of colorectal tumor whole-genomes reported to date, and identified significant associations with clinical characteristics.

**Table 2 Multiple linear regression model for log insertion counts**

|  | Coefficient | Std. err. | z | p | Signif. |
|---|---|---|---|---|---|
| Intercept | 0.408 | 0.647 | 0.630 | 5.29e-01 |  |
| CIMP-H | 0.607 | 0.169 | 3.60 | 3.22e-04 | *** |
| Allelic Imbalance (/10% of reference) | 0.0826 | 0.0284 | 2.91 | 3.64e-03 | ** |
| TP53 mutation | −0.0684 | 0.134 | −0.509 | 6.10e-01 |  |
| MSI | 0.150 | 0.285 | 0.527 | 5.98e-01 |  |
| Mean coverage (/10 reads) | 0.309 | 0.0862 | 3.59 | 3.33e-04 | *** |
| Age at diagnosis (/10 years) | 0.0331 | 0.0603 | 0.549 | 5.83e-01 |  |
| Male | 0.0570 | 0.123 | 0.464 | 6.42e-01 |  |
| Dukes B | −0.0578 | 0.168 | −0.344 | 7.31e-01 |  |
| Dukes C | −0.0483 | 0.186 | −0.259 | 7.96e-01 |  |
| Dukes D | 0.00490 | 0.211 | 0.0232 | 9.81e-01 |  |
| Proximal location | 0.206 | 0.144 | 1.43 | 1.52e-01 |  |

*MSI* microsatellite instability, *CIMP-H* CpG methylator phenotype high
Significance codes: *** ≤ 0.001 < ** ≤ 0.01 < * ≤ 0.05 < . ≤ 0.1

**Table 3 Cox proportional hazards model for disease-specific survival**

|  | Coefficient | Std. err. | z | p | HR [95% CI] | Signif. |
|---|---|---|---|---|---|---|
| Insertion count (/10) | 0.108 | 0.0362 | 2.98 | 2.93e-03 | 1.11 [1.04, 1.20] | ** |
| MSI | −0.258 | 0.642 | −0.402 | 6.88e-01 | 0.773 [0.219, 2.72] |  |
| CIMP-H | 0.174 | 0.341 | 0.510 | 6.10e-01 | 1.19 [0.610, 2.32] |  |
| BRAF mutation | 0.790 | 0.447 | 1.77 | 7.68e-02 | 2.20 [0.918, 5.29] | . |
| Age [55, 75) years | −0.147 | 0.408 | −0.360 | 7.19e-01 | 0.863 [0.388, 1.92] |  |
| Age ≥ 75 years | 0.188 | 0.427 | 0.439 | 6.60e-01 | 1.21 [0.523, 2.78] |  |
| Male | 0.311 | 0.232 | 1.34 | 1.80e-01 | 1.37 [0.866, 2.15] |  |
| Dukes B | 0.452 | 0.449 | 1.01 | 3.13e-01 | 1.57 [0.652, 3.79] |  |
| Dukes C | 1.77 | 0.431 | 4.12 | 3.82e-05 | 5.89 [2.53, 13.7] | *** |
| Dukes D | 2.78 | 0.454 | 6.12 | 9.07e-10 | 16.2 [6.64, 39.4] | *** |
| Allelic Imbalance (/10% of reference) | −0.0583 | 0.0539 | −1.08 | 2.80e-01 | 0.943 [0.849, 1.05] |  |

The model was stratified by tumor location
*HR* Hazard ratio, *CI* confidence interval, *MSI* microsatellite instability, *CIMP-H* CpG island methylator phenotype high
Significance codes: *** ≤ 0.001 < ** ≤ 0.01 < * ≤ 0.05 < . ≤ 0.1

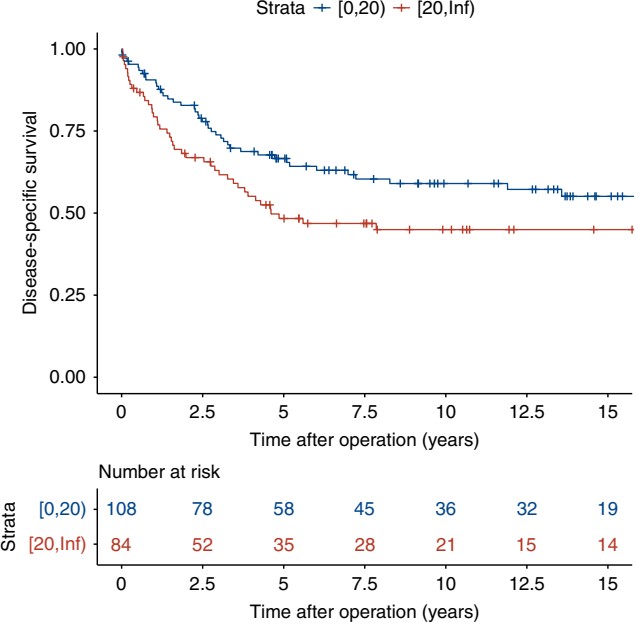

**Fig. 6** Kaplan–Meier curves by insertion count. Tumors with less than 20 somatic insertions (blue line) and tumors with 20 or more insertions (red line)

We observed high retrotransposon activity with wide variability among tumors. We confirmed higher insertion density in late replicating regions, closed chromatin. Among insertions in genes, we also observed higher insertion count in genes with lower expression. The list of the most active reference LINE-1s became also validated in this extended set of CRCs[10,11,21].

A number of additional observations were made. We identified recurrent insertions in 333 protein-coding genes, 15 of which are included in the CGC[29]. The most recurrent hit was *LRP1B* with 19 intronic insertions. The high frequency of insertions in this gene could be a result of various characteristics such as chromatin state, replication timing as well as gene length and/or expression. However, other causes such as sequence composition or somatic selection cannot be excluded. We also observed a high frequency of insertions at fragile sites with lower gene expression and lower AI fraction. These findings suggest that while AI is a recurrent feature of some fragile sites, sites with lower gene expression are more prone to retrotransposon insertions in CRC. Yet, the molecular basis of frequent retrotransposition in fragile sites, in particular *LRP1B*, and whether these insertions are important for the tumorigenic process remain as open questions.

Among the exonic insertions, we identified one in *PIK3CA* and two in *APC*. *PIK3CA* is a known oncogene involved in colorectal tumor progression and mutations in *APC* lead to colorectal tumor initiation[33,34]. The insertion locations in *APC* were similar to two previously reported insertions[24,25] and consistent with the distribution of pathogenic somatic changes in this gene. Similar to most pathogenic *APC* mutations, the insertions were predicted to disrupt the open reading frame of the gene. These observations, together with the previous extensive work showing the key role of *APC* loss early in colorectal neoplasia[33,34], suggest that retrotransposon insertions in *APC* is one mechanism of CRC initiation as previously proposed[24,25], although the inactivation of *APC* by retrotransposition should be functionally assessed in future studies. In addition, we identified recurrent intronic retrotranspositions in other genes frequently mutated in CRC, such as *FAT4*, and whether these insertions are important for the tumorigenic process remains to be investigated.

The availability of patient data allowed us to investigate possible associations between somatic insertion count and various molecular and clinical characteristics. We applied a multiple linear regression model and found that retrotransposon activity was positively associated with the genomic fraction of AI, and paradoxically with CIMP even though LINE-1s are frequently repressed by promoter methylation[12]. Moreover, LINE-1 methylation was associated with MSI and the CIMP in a previous study in CRC[37]. Of note, both CIMP and the genomic fraction of AI are characteristic of the two distinct genetic instability pathways in CRC. No associations with age at diagnosis, *TP53* mutations or tumor stage were detected in contrast to other studies[18,38]. These studies had significantly smaller sample sizes and fewer covariables were taken into account, which may explain the discrepancy. Importantly, survival analysis revealed a significant association between insertion count and poor disease-specific survival independently of other prognostic factors. Our findings indicate that tumors with high retrotransposon activity present characteristics of both MSS and MSI tumors, and are associated with poor CRC-specific survival. Although, further studies need to confirm the prognostic value of retrotransposon insertions not only in CRC, but also in other cancer types.

By characterizing the landscape of retrotransposon insertions in a large dataset of CRCs, we found that retrotranspositions appear to have the ability to serve as tumor-initiating events in CRC. The association of retrotransposition events with clinical characteristics—in particular poor prognosis—suggest that retrotransposition may play a more important role in CRC than previously thought. Further work should elucidate the timing and mechanisms leading to high somatic retrotransposition activity in some individuals, while others are spared. Understanding these could provide tools for management of CRC, including prevention.

## Methods

**Study subjects**. The samples and the clinical data utilized in this study were obtained from a population based series of 1042 CRCs[39,40] and from a subsequently collected series of additional Finnish CRCs. The tumors were fresh frozen and the corresponding normal tissues were obtained from either blood or from the normal colon tissue. Originally 202 CRCs entered the analyses. However, one tumor was later classified as an advanced adenomatous lesion (c232.1T). All samples were collected after informed consent. In the great majority of cases the consent was signed, in few cases collected before 1999 a verbal consent was derived (signed informed consent was not required in the Finnish legislation prior to that). For these early samples subsequent authorization for research use was derived from the National Supervisory Authority for Welfare and Health (Dnro 421/04/044/06, Dnro 8048/06.01.03.01/2014, Dnro 358/32/300/05, Dnro 1476/06.01.03.01/2012). The study has been reviewed by the Ethics Committee of the Hospital district of Helsinki and Uusima (Dnro 133/E8/03, 408/13/03/03/2009). Permission to use patient information was obtained from the National Institute for Health and Welfare (Dnro 53/07/2000, Dnro THL/1071/5.05.00/2011, Dnro THL/151/5.05.00/2017).

**Whole genome sequencing**. WGS was performed on Illumina HiSeq 2000 with 100 bp paired-end reads. Each normal and tumor DNA was sequenced to at least 40× median coverage. Data was processed similar to GATK best practices[36,41].

**Transposon detection**. The identification of somatic retrotransposon insertions was conducted utilizing the Transposon Finder in Cancer (TraFiC)[10]. TraFiC default parameters were applied except for; $a = 1$ (RepeatMasker accuracy), $s = 3$ (minimum of three reads in tumor cluster), and $gm = 3$ (minimum of three reads in normal cluster). In addition, paired-end reads with both ends having equal mapping quality and above 0 were included. In these cases, the first end of the pair was selected as the anchor read (end mapping to non-repetitive sequence). RepeatMasker (version open-4.0.5) and NCBI/RMBLAST 2.2.27+ were used for retrotransposon alignment as part of TraFiC. The RepeatMasker Database release utilized in this study was 20140121[42,43]. Somatic filtering was performed against germline calls from 234 normal samples (202 corresponding CRC normals, 20 myometrium samples[44], and 12 blood samples[45]) with a 200 bp window as described in TraFiC[10]. Furthermore, calls in decoy sequences from 1000 Genomes Project Phase 2 (hs37d5)[46] were filtered away.

**Detection of LINE-1 transductions**. We identified 3′ and orphan transductions utilizing DELLY structural variant (SV) calls (v 0.0.9)[28,41]. Filtering criteria utilized in this study were: SV calls supported by at least three supporting discordant reads and mapping quality > 37. SV calls were merged if they were the same DELLY type and were within 200 bp. Merged SVs (SV length > 1000 bp) in tumors were filtered against merged SVs in the normal samples. Subsequently, we extracted the SVs with one end of the pair within 1000 bp from the 3′ end of a reference human-specific LINE-1 (Reference L1HS, full-length) from The European database of L1-HS retrotransposon insertions in humans (euL1db)[47], database version v1.0, date 05-10-14. The other end of the pair was used in the somatic filtering, where a 200 bp window and transduction calls from the pool of normal samples above mentioned were applied. One transduction detected in a female coming from an LINE-1 in Yp11.2 was filtered away. Furthermore, transduction calls within 200 bp, from the same retrotransposon family, and in the same sample were regarded as the same insertion and merged together. The same rationale was applied for calls detected by both DELLY and TraFiC.

**Methylation**. A Methylation-Specific Multiplex Ligation-dependent Probe Amplification assay (MS-MLPA) (Nygren AO, 2005) with the SALSA MLPA ME042 CIMP probemix (MRC-Holland, Amsterdam, The Netherlands) was used to determine the CIMP in an extended set of 255 tumor samples and the corresponding normal colon tissue of 175 samples as a separate study. Data from normal samples were used to determine the threshold for hypermethylation in the tumor samples. MS-MLPA was performed according to manufacturer's instructions[48] (http://www.mrc-holland.com Accessed December 2015). In short, the assay targets the promoter region of eight tumor suppressor genes; *CACNA1G, CDKN2A, CRABP1, IGF2, MLH1, NEUROG1, RUNX3, SOCS1*. The methylation level for each probe was called using the Coffalyser software (MRC-Holland, Amsterdam, The Netherlands). If ≥25% of the probes for one gene were methylated, the gene was scored as methylated. If 5–8 genes, were scored as methylated, the tumor was classified as CIMP-high (CIMP-H), and if 0–4 genes were scored as methylated it was classified as CIMP-low (CIMP-L) tumor (Source data are provided as a Source Data file).

**RNA sequencing**. Total RNA from consecutive cryosections was extracted using RNeasy Mini Kit (Qiagen) from 34 tumors that displayed more than 50% of cancer cell percentage (HE staining of cryosections) and RNA integrity > 6 (Agilent RNA 6000, Agilent 2100 Bioanalyzer). Paired-end RNA sequencing was performed on the Illumina Hiseq 2000[49]. RNA-seq data was processed using Kallisto (version 0.43.0) software[50]. Kallisto quantification was executed in paired-end mode and aligned against the Ensembl Human reference transcriptome (GRCh37_79). Quantification results from Kallisto were normalized and aggregated to gene-level utilizing sleuth (version 0.28.1) R package with default filtering settings[51].

**Visual inspection of paired-end read data**. We selected 100 random insertions to ascertain the rate of true somatic calls based on visual inspection of the paired-end read data. Visualization was performed with BasePlayer[52]. Somatic calls were visually validated as true if the insertion call was supported by discordant reads (three + three for TraFiC calls) and at least two split reads supporting the insertion breakpoint and/or the polyA/T. Furthermore, the corresponding normal tissue was also visualized to confirm the somatic origin of the insertion calls.

**Insertion annotation**. Annotation of the insertion calls was applied by using the inner genomic coordinates of the reciprocal clusters provided by TraFiC (P_R_POS & N_L_POS) (Supplementary Data 1). Insertion breakpoints hitting an intron or an exon of any protein-coding transcript (GRCh37_87) were annotated as intron/exon hit. Insertion orientation was determined by the presence of a polyA or a polyT (within a 200 bp window from mid point between positive breakpoint and negative breakpoint) in conjunction with gene orientation. PolyA was called when at least two forward strand reads started with three or more consecutive "A" bases. We used sequences of other reads to detect "A" repeats in the reference (i.e., the polyA call was discarded if "A" repeat was found in the middle of other overlapping read). PolyT was called using the reverse strand reads with three or more consecutive "T" bases at the end of the read sequence not present in the reference as described above (Supplementary Data 1). Insertion strand with respect to reference was defined as reverse when a polyA was called, and defined as forward when a polyT was called. Sense insertions were defined when the insertion strand with respect to reference was reverse in genes in plus orientation or forward in genes in minus orientation. Antisense insertions were defined when insertion strand with respect to reference was reverse in genes in minus orientation or forward in genes in plus orientation (Supplementary Data 1). Replication time fractions were extracted from Chen et al.[53]. Insertion density was defined as number of insertions divided by the total number of base pairs of each replication time fraction. Open chromatin was defined as DNAse regions that were overlapping in at least two out of the four cell lines (RKO, LoVo, CaCo2, and Gp5D) (GSE83968)[54] in the 1000 Genomes Project pilot style callable regions[55]. Closed chromatin regions were defined as the above-mentioned callable regions minus the open chromatin regions.

**GO analysis**. We applied GO analyser for RNA-seq and other length biased data (Goseq)[56] with R version 3.5.1 to identify enrichment of biological processes in genes with recurrent insertions. Genes with recurrent insertions were defined as genes with two or more insertions. We utilized Wallenius approximation and p-values were corrected using Benjamini and Hochberg method. Enrichment was considered for FDR corrected p-values above 0.05.

**Fragile sites**. The 21 fragile sites were defined as genes with more than 0.85 probability of being fragile (Random forest 3 predictors)[30]. Genomic coordinates were lifted to GRCh37/Hg19 with https://genome.ucsc.edu/cgi-bin/hgLiftOver[57] and regions with no converted coordinates were excluded (chr10:46597226–48877831, chr10:45970128–48447930). The fraction of AI was calculated as the number of focal AI events per fragile site (both breakpoints of each AI call within the fragile site coordinates) divided by the total number of AI events in 1699 tumors[36]. Insertion fraction was calculated as the number of insertions per fragile site divided by the total number of insertions detected in 202 patients. Fragile site categories were defined based on the ratio of insertion fraction/AI fraction. AI high; 0 < ratio < 1, and Retrotransposon-high; ratio > 1.

**Mutation analysis in CRC genes**. Somatic changes in *BRAF, KRAS, TP53* and *APC* were called using MuTect (version 1.1.4) with default parameters (GRCh37_78)[36,41]. Subsequent filtering criteria were minimum coverage of 4, minimal allelic fraction of 10 and minimum quality score 20[52]. For *KRAS*, mutations in codons 12, 13, 61, 117, and 146 in any transcript were classified as mutation positive and for *BRAF*, only hotspots in V600E in any transcript were considered as mutation positive. All non-synonymous changes in any transcript of *TP53* were classified as mutation positive. In addition, non-synonymous changes in *APC* (ENST00000457016) from 234 MSS tumors[36] were utilized for Fig. 4. Figure 4 was created with http://www.cbioportal.org/tools.jsp[58,59] and modified with Inkscape (http://www.inkscape.org)[60].

**Association test between insertions and RNA expression**. For the 827 insertions identified in any of the 34 tumors, we investigated the effect on the expression of the 642 distinct closest genes. For each sample and each gene the TPM values were extracted and ranked in ascending order. Consequently, the rank number corresponding to the sample with the insertion was recorded for each gene. We computed the sum-of-squared error statistic (Chi-square test) for the frequency table to test whether the rank values of the samples with insertion were uniformly distributed (no insertion effect on gene expression). Furthermore, 100,000 permutations with randomized rank numbers were applied but no significant effect was observed (Supplementary Fig. 2). Tests were performed using R versions 3.4.3 or 3.3.0.

**Multiple linear regression analysis**. To model retrotransposon insertion counts we applied a multiple linear regression model for log-transformed insertion counts. Spearman correlation matrix (R package PerformanceAnalytics) and variance inflation factors (vif function in R package car) were computed to evaluate possible collinearity among explanatory variables[61,62]. Model fit was assessed by plotting residuals against fitted values, theoretical normal quantiles and leverage (Supplementary Figs. 3–5). All tests were performed using R version 3.3.2[63].

**Cox proportional hazards regression analysis**. We applied the Cox proportional hazards regression to study the association between disease-specific survival with retrotransposon insertion counts. The time variable was defined as days since diagnosis or operation. Patients that were alive in the last status assessment were censored at that date (survival status was assessed periodically using the Population Register Centre of Finland with the most recent assessment in 2016). Death from other causes than CRC were also defined as censored events. Proportional hazards assumptions were assessed by Grambsch-Therneau test for proportional hazards and evaluation for a non-zero slope of the scaled Schoenfeld residuals versus time (Supplementary Fig. 6). Based on inspection of the scaled Schoenfeld residuals, the model was stratified by tumor location. Influential observations were assessed with dfbeta and martingale residuals (Supplementary Figs. 7 and 8). All tests were performed using R version 3.3.2[63].

**Reporting summary**. Further information on research design is available in the Nature Research Reporting Summary linked to this article.

## Data availability

The raw sequencing data produced in this study is not available due to the presence of germline data- and thus identifiable information-which we do not have the specific consent to distribute. The whole-genome somatic point mutations have been deposited in the EGA database under the accession code EGAS00001003010. Gene expression values have been deposited in the Zenodo database under the Digital Object Identifier [https://doi.org/10.5281/zenodo.3241399]. The methylation source data and the data underlying Fig. 3 are provided as Source Data files. All the other data supporting the findings of this study are available within the article and its Supplementary Information

files. A reporting summary of this article is available as a Supplementary Information files.

## Code availability

The code is available in https://github.com/cajuso/RetroCRCmanu

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

## Acknowledgements

The authors thank Alison Ollikainen, Iina Vuoristo, Inga-Lill Åberg, Sini Marttinen, Marjo Rajalaakso, Sirpa Soisalo, Jiri Hamberg, and Heikki Metsola for technical assistance and Alison Ollikainen also for proofreading the manuscript. This work was supported by grants from the Academy of Finland (Finnish Center of Excellence Program 2012–2017, 250345 and 2018–2025, 312041), The Finnish Cancer Society, The European Research Council (268648), The Sigrid Juselius Foundation; Jane and Aatos Erkko Foundation, the Nordic Information for Action eScience Center (NIASC) and Nordic Center of Excellence financed by NordForsk (Project number 62721). We also thank SYSCOL (an EU FP7 Collaborative Project, 258236) for sequencing the RNA samples. The following foundations are acknowledged for personal funding: Ida Montinin Säätio foundation, Cancer Society of Finland, Juhani Ahon Foundation for Medical Research and The Maud Kuistila Memorial Foundation. The authors wish to acknowledge CSC-IT Center for Science, Finland, for computational resources.

## Author contributions

T.C., P.S., and T.T. analyzed insertion data. T.C., P.S. and E.P. contributed and performed insertion and transduction calling. T.C., T.T., U.A.H. and J.K. prepared WGS samples. O.K. supervised WGS sample preparation. T.C. and O.K. contributed and organized RNA sample preparation. T.C. and T.T. performed statistical analysis. R.K., E. P., K.P. and N.V. were involved in primary WGS data analysis. T.C. and R.K. performed and analyzed insertion polyA/T calls and insertion orientation bias, and R.K. developed BasePlayer. A.T. and L.F. performed CIMP analysis. A.T. and K.P. designed and performed the analysis of insertion effect on gene expression. N.V. performed primary RNA-seq analysis. A.R. and J.B. reviewed tumors. S.K., A.L., L.R.S., T.K., T.S. and J.P.M. provided patient samples. E.K. and M.A. contributed to the study design. T.C., O.K., E.P., K.P. and L.A.A. designed the study. O.K., E.P., K.P. and L.A.A. supervised the study. All authors contributed to writing the manuscript.
