## [Peer Review File · Nature Communications]

Reviewers' comments:

Reviewer #1 (Remarks to the Author):

The article, Retrotransposon insertions can initiate colorectal cancer and are associated with poor survival, by Cajuso and colleagues, reviews more than 200 colorectal cancer genomes to relate LINE-1 retrotransposition to clinical features. They find 1% of cases have tumor 'driving' insertions in APC, and associate L1 activity with the CpG island methylator phenotype, allelic imbalance, and poor survival.

Overall, many of these are known activities or associations of LINE-1 in this disease. However, the study is valuable in that it demonstrates the incidence of strength of associations in a large cohort. The survival analysis in particular has not been conducted previously with covariate corrections as applied here on this number of samples to my knowledge. It is a nice addition to the field, and should be of interest to the readership of Nature Communications.

1. A brief introduction to TraFiC and DELLY for detecting retrotranspositions should be mentioned in the results. Information comparing the performance of these pipelines would be a nice addition.
2. Figure 1B – provide units for the axes
3. Figure 2B – A correction should be made for gene size before considering enrichment of biological process.
4. A more comprehensive analysis of genes recurrently affected by L1 insertions would be a good addition (i.e., instead of limiting to CGC genes). After a correction for gene size, what genes are recurrently involved?
5. For LRP1B and other genes with significant numbers of insertions, consider providing more information about insertion locations. Do the 19 insertions in LRP1B cluster in the same intron? How do they relate to known fragile sites in the locus? What is the orientation bias considering this set of 19?
6. Presumably the insertions in ex16 of APC interrupt the protein reading frame, but this should be stated.
7. There is a misspelling on the Figure 3 y axis label.
8. More information about how fragile site intervals were defined would be useful in the Results.
9. More information on active elements would be helpful in the text (rather than only appearing in table S4). How many of the 56 source L1s identified by 3' transductions known to be active for retrotransposition? Are they all full-length insertions with intact ORFs? Did the authors identify new elements?
10. Was the study of source elements limited to elements in the reference genome? (All seem about 6kb in size based on genome coordinates in the supplementary table.) What about the many, active polymorphic L1s?
11. For the CIMP and genomic fraction of allelic imbalance, consider providing more information than just the association in Table 2. These are interesting relationships, and a dedicated figure would be a nice addition.
12. Additional citations would be helpful to add. For example, PMID 28642606 as a review of LINE-1 in cancer; PMID 18366060 in considering CIMP and LINE-1 methylation.

Reviewer #2 (Remarks to the Author):

In the present manuscript, Cajuso and colleagues mapped retrotransposon insertions in colorectal cancer by comparing whole genome sequencing data from 202 colorectal tumors with their matched normal samples. The authors integrated such data with RNA-seq for a small subset of samples and also collected relevant clinical information. The authors came across several observations such as (i) retrotransposon insertions took place in the sites of cancer genes, including APC; (ii) recurrent retrotransposon insertions happened at sites of fragile sites; (iii) retrotransposon insertion was significantly associated with CIMP phenotype and with the fraction of

allelic imbalance; and (iv) insertion count was associated with poor disease-specific survival in a multivariate test. Overall, the manuscript sheds light into the role of retrotransposons in human carcinogenesis and points to an unprecedented significance of this phenomenon in clinical relevant parameters. Nevertheless, the study would require some clarifications:

- 1- According to their analysis, LRP1B is the most frequently affected protein-coding gene with retrotransposon insertions. However, the authors already acknowledge that LRP1B is a known fragile site. In addition, this is one of the largest gene in the human genome, so it is not clear to this reviewer the biological impact of these associations.
- 2- Figure 2b shows that most enriched biological processes are neuron-neuron synaptic transmission and cell-cell adhesion, any of which seem to have a close relationship with initializing colorectal tumorigenesis.
- 3- Inactivation of APC through retrotransposon insertion might require of functional validation in order to proof involvement in tumor initiation.
- 4- The negative correlation between retrotransposon insertion and gene expression is restricted to 34 tumors. Despite the fact that a statistical significance is shown, it would be very interesting to provide data for the 202 colorectal tumors and approach whether this correlation is also maintained in genomic regions other than fragile sites.
- 5- The prognostic value of retrotransposon insertion should definitely be validated in an independent cohort.
- 6- The association between retrotransposon insertion and allelic imbalance requires a further detailed exploration and in depth analysis. Are the insertions at site of chromosome breakpoints?
- 7- Overall, the observations the authors made, although some are novel and intriguing, do not seem to be decisive to make some strong statements (e.g., <<retrotransposon insertion in APC is one mechanism of CRC initiation>>).

Reviewers' comments:

Reviewer #1 (Remarks to the Author):

The article, Retrotransposon insertions can initiate colorectal cancer and are associated with poor survival, by Cajuso and colleagues, reviews more than 200 colorectal cancer genomes to relate LINE-1 retrotransposition to clinical features. They find 1% of cases have tumor 'driving' insertions in APC, and associate L1 activity with the CpG island methylator phenotype, allelic imbalance, and poor survival.

Overall, many of these are known activities or associations of LINE-1 in this disease. However, the study is valuable in that it demonstrates the incidence of strength of associations in a large cohort. The survival analysis in particular has not been conducted previously with covariate corrections as applied here on this number of samples to my knowledge. It is a nice addition to the field, and should be of interest to the readership of Nature Communications.

1. A brief introduction to TraFiC and DELLY for detecting retrotranspositions should be mentioned in the results. Information comparing the performance of these pipelines would be a nice addition.

We have added a short introduction to both methods with the number of insertions detected by each method. This can now be found at the beginning of the results section "**Genome-wide detection of somatic retrotransposition in CRC**", page 2.

2. Figure 1B – provide units for the axes

We have now improved Figure 1B following the reviewer's suggestion. We have added the units in X-axis, "Replication time fraction"; and Y-axis, "Insertion density per tumor/bp".

3. Figure 2B – A correction should be made for gene size before considering enrichment of biological process.

We followed the reviewer's suggestion and repeated the analysis. We have now used Gene Ontology analyser for RNA-seq and other length biased data (Goseq). Briefly, genes with recurrent insertions were defined as genes with two or more insertions and Wallenius approximation was applied to estimate the probability of a gene having recurrent insertions after adjusting for gene length. After correcting for gene length and multiple testing, we did not observe any significant enrichments. The results have now been updated at the end of the first paragraph of the results section "**Retrotranspositions are predicted to initiate ~ 1% of CRCs**", page 4. The methods section "**Gene ontology analysis**", page 18 has now been updated with the description of Goseq.

Previous results that did not include gene length correction have now been removed from the manuscript.

4. A more comprehensive analysis of genes recurrently affected by L1 insertions would be a good addition (i.e., instead of limiting to CGC genes). After a correction for gene size, what genes are recurrently involved?

We appreciate the reviewer's comment and we have now added a column with insertion density per gene in Supplementary Table 2. Genes with the highest density of recurrent insertions were *RCNI* with 3 insertions, and *COL25A1*, *ARAP2* and *ZNF251* with 2 insertions. A paragraph with extended information about these genes has now been added in the results section "**Retrotranspositions are predicted to initiate ~ 1% of CRCs**", page 6.

5. For *LRP1B* and other genes with significant numbers of insertions, consider providing more information about insertion locations. Do the 19 insertions in *LRP1B* cluster in the same intron? How do they relate to known fragile sites in the locus? What is the orientation bias considering this set of 19?

We agree with the reviewer's comment and we have expanded the results section "**Retrotranspositions are predicted to initiate ~ 1% of CRCs**", page 5-6. In addition, we have added a new figure (**Figure 3**) showing the location of retrotransposon insertions identified in this study as well as human papillomavirus integration breakpoints (Hu et al. 2015), and allelic imbalance breakpoints (Palin et al. 2018) for both *LRP1B* and *DLG2*.

6. Presumably the insertions in ex16 of *APC* interrupt the protein reading frame, but this should be stated.

We have included a sentence stating that the insertions in *APC* are predicted to disrupt the protein as it was previously reported in two other studies (Scott et al. 2016; Miki et al. 1992).

7. There is a misspelling on the Figure 3 y axis label.

Thanks for spotting this; we have now corrected the misspelling in the figure.

8. More information about how fragile site intervals were defined would be useful in the Results.

We have updated the results section "**Recurrent insertions in lowly expressed fragile sites**", page 8 with the reviewer's suggestions. We have included a sentence explaining the selection criteria of fragile sites and a sentence clarifying the classification of fragile sites based on insertion fraction. In addition, we have changed the Y-axis category names in Figure 4b to make the classification clearer to the reader.

9. More information on active elements would be helpful in the text (rather than only appearing in table S4). How many of the 56 source L1s identified by 3' transductions known to be active for retrotransposition? Are they all full-length insertions with intact ORFs? Did the authors identify new elements?

We agree with the reviewer's comment and we have now added how many source LINE-1s were previously reported to be active in humans (Brouha et al. 2003; Beck et al. 2010) as well as how many were reported to be active in cancer (Rodriguez-Martin et al. 2017) (results section "**Few active reference LINE-1s account for most transductions**", page 9).

We only selected elements that were full-length as defined by the integrity tab in the reference file (<http://eul1db.unice.fr/Data.jsp>). From the database, 313 out of 315 full-length reference LINE-1s were at least 6,000 bp long. There were two elements that were 3,619 and 5,770 bp long (chr1:69202376-69205819 and chr8:105971290-105977088, respectively) however no somatic activity form either was detected. We did not detect any novel active full-length elements since our study was limited to elements in the reference genome. This limitation has now been highlighted in the results section "**Few active reference LINE-1s account for most transductions**", page 9.

10. Was the study of source elements limited to elements in the reference genome? (All seem about 6kb in size based on genome coordinates in the supplementary table.) What about the many, active polymorphic L1s?

This is correct, the detection of 3' transductions was limited to only full-length elements present in the human reference genome. Consequently, we could not detect the activity or presence of novel polymorphic elements. We acknowledged the limitation pointed out by the reviewer and decided to clarify this limitation by changing "LINE-1" to "reference LINE-1" in the title and beginning of the results section "**Few active reference LINE-1s account for most transductions**", page 9 as well as in the discussion, page 12. In addition, we added a sentence at the end of the results section "**Few active reference LINE-1s account for most transductions**", page 9 stating the limitation.

11. For the CIMP and genomic fraction of allelic imbalance, consider providing more information than just the association in Table 2. These are interesting relationships, and a dedicated figure would be a nice addition.

Inspired by the reviewer's suggestion, we explored these associations in more depth. We further investigated the association with CIMP by repeating the multiple linear model including *BRAF* mutation (V600E) as a variate, and only including MSS samples. Furthermore, we investigated whether insertion breakpoints and allelic imbalance breakpoints were colocalizing events. We have now updated the results section "**Insertion count associates with CIMP and allelic imbalance**", page 9-10. In addition, we have added a new figure (**Figure 3**) with a gene map of the insertion breakpoints and allelic imbalance breakpoints in both genes with highest insertion count, *LRP1B* and *DLG2*.

12. Additional citations would be helpful to add. For example, PMID 28642606 as a review of LINE-1 in cancer; PMID 18366060 in considering CIMP and LINE-1 methylation.

We thank the reviewer for this suggestions and both citations have now been included in the manuscript.

PMID 28642606 has been included at the end of the sentence “*LINE-1s are frequently repressed by promoter methylation (Yoder et al. 1997) and genome-wide hypomethylation is reported to lead to their activation during tumorigenesis (Alves et al. 1996, Shukla et al. 2013), thus leading to high retrotransposon activity and genome instability (Shukla et al. 2013, Daskalos et al. 2009, Burns 2017)*” in the introduction, page 1.

In addition, we have included a sentence in the discussion, page 12 containing the suggested citation (PMID 18366060), “*Moreover, LINE-1 hypomethylation was inversely associated with microsatellite instability and the CpG island methylator phenotype in a previous study in CRC (Ogino et al. 2008).*”

Reviewer #2 (Remarks to the Author):

In the present manuscript, Cajuso and colleagues mapped retrotransposon insertions in colorectal cancer by comparing whole genome sequencing data from 202 colorectal tumors with their matched normal samples. The authors integrated such data with RNA-seq for a small subset of samples and also collected relevant clinical information. The authors came across several observations such as (i) retrotransposon insertions took place in the sites of cancer genes, including APC; (ii) recurrent retrotransposon insertions happened at sites of fragile sites; (iii) retrotransposon insertion was significantly associated with CIMP phenotype and with the fraction of allelic imbalance; and (iv) insertion count was associated with poor disease-specific survival in a multivariate test. Overall, the manuscript sheds light into the role of retrotransposons in human carcinogenesis and points to an unprecedented significance of this phenomenon in clinical relevant parameters. Nevertheless, the study would require some clarifications:

1- According to their analysis, LRP1B is the most frequently affected protein-coding gene with retrotransposon insertions. However, the authors already acknowledge that LRP1B is a known fragile site. In addition, this is one of the largest gene in the human genome, so it is not clear to this reviewer the biological impact of these associations.

We agree with the reviewer’s comment, the impact of retrotransposon insertions in *LRP1B* is unclear. In fact, *LRP1B* has also been identified as a recurrent hotspot for human papillomavirus integrations in cervical cancers which has now been added to the results section “**Retrotranspositions are predicted to initiate ~ 1% of CRCs**”, page 4. We also highlight the uncertainty in the discussion, page 12-13 “*The high frequency of insertions in this gene could be a result of various characteristics such as chromatin state, replication timing as well as gene length and/or expression. However, other causes such as sequence composition or somatic selection cannot be excluded*”.

2- Figure 2b shows that most enriched biological processes are neuron-neuron synaptic transmission and cell-cell adhesion, any of which seem to have a close relationship with initializing colorectal tumorigenesis.

We have now applied a new tool, Gene Ontology analyser for RNA-seq and other length biased data (Goseq). After adjusting for gene length no biological processes were significantly enriched. The results section has been updated accordingly (results section “**Retrotranspositions are predicted to initiate ~ 1% of CRCs**”, page 4. And methods section “**Gene ontology analysis**”, page 18).

3- Inactivation of APC through retrotransposon insertion might require of functional validation in order to proof involvement in tumor initiation.

We understand the reviewer’s comment. However, *APC* is the well-established gatekeeper of the colonic crypt homeostasis and lost at colorectal tumor initiation in a striking proportion of colorectal neoplasia cases, as shown in dozens or hundreds of works since 1991 (Vogelstein et al. 2013; Fearon 2011). Since both insertions were identified in the coding region of *APC*, they were predicted to be events that disrupt the normal *APC* protein (results section “**Retrotranspositions are predicted to initiate ~ 1% of CRCs**”, page 7). Two other similar cases have been previously reported in the literature (Scott et al. 2016; Miki et al. 1992). Taken together the data by us and others, it is not unreasonable to conclude that the insertions identified in this study have had the potential to contribute to tumor initiation in the patients. However, we have now toned down the manuscript in both the results title and section, as well as in the discussion, and think that the results and conclusions are now well balanced. As always, we are of course happy to hear any additional editorial advice.

4- The negative correlation between retrotransposon insertion and gene expression is restricted to 34 tumors. Despite the fact that a statistical significance is shown, it would be very interesting to provide data for the 202 colorectal tumors and approach whether this correlation is also maintained in genomic regions other than fragile sites.

The RNA data was restricted to the 34 tumors where data is available for us, as the reviewer correctly indicates. However, it was used in three different analysis in the manuscript. First, in the results section “**Retrotranspositions are predicted to initiate ~ 1% of CRCs**”, page 4. and in **Figure 2** where we observed that insertion count was higher in genes with lower expression than genes with higher expression. In this case, we utilized expression data for all the protein-coding genes used for insertion annotation. Second, at the end of the results section “**Retrotranspositions are predicted to initiate ~ 1% of CRCs**”, page 7, we utilized all genes in close proximity to a retrotransposon insertion in any of the 34 tumors to investigate whether insertions affected gene expression (n=642 protein-coding genes). Third, in the results section “**Recurrent insertions in lowly expressed fragile sites**”, page 8, we investigated the frequency of insertions and allelic imbalances related to gene expression in

fragile sites. This analysis was limited to only fragile sites due to the high frequency of insertions and focal allelic imbalances in these genomic regions.

5- The prognostic value of retrotransposon insertion should definitely be validated in an independent cohort.

We agree with the reviewer's comment; unfortunately we have found that WGS data - necessary for the effort - is sparsely available from CRC. We tried to validate the prognostic value of retrotransposon insertions by repeating the survival analysis utilizing the ICGC PCAWG CRC cohort (n=62). However, follow-up time was only 36 person-years in contrast to our study where it was 1,370 person-years. This data was not sufficient for a meaningful analysis. However, we agree that this is an important point and added the following sentence to highlight the need for future validation to the readers. Discussion, page 14 "*Importantly, survival analysis revealed a significant association between insertion count and poor disease-specific survival independently of other prognostic factors, although, the prognostic value of retrotransposon insertions must be validated in an independent cohort as one becomes available.*"

6- The association between retrotransposon insertion and allelic imbalance requires a further detailed exploration and in depth analysis. Are the insertions at site of chromosome breakpoints?

Following the reviewer's suggestion we investigated whether insertions colocalized with chromosomal allelic imbalance breakpoints. We only identified one colocalizing event in one sample. This information and a more extended exploration of these associations has now been added at the end of the results section "**Insertion count associates with CIMP and allelic imbalance**" page 9.

7- Overall, the observations the authors made, although some are novel and intriguing, do not seem to be decisive to make some strong statements (e.g., <<retrotransposon insertion in APC is one mechanism of CRC initiation>>).

We agree with the reviewer's comment and we have now toned down the manuscript in both the results title and section, as well as in the discussion. We explain reiterate here our reasoning in this matter: *APC* is the well-established gatekeeper of the colonic crypt homeostasis and lost at colorectal tumor initiation in a striking proportion of colorectal neoplasia cases, as shown in dozens or hundreds of works since 1991 (Vogelstein et al. 2013; Fearon 2011). Since both insertions were identified in the coding region of *APC*, they were predicted to be events that disrupt the normal *APC* protein (results section "**Retrotranspositions are predicted to initiate ~ 1% of CRCs**", page 7). Two other similar cases have been previously reported in the literature (Scott et al. 2016; Miki et al. 1992). Thus, taken together the data by us and others, it is not unreasonable to conclude that the insertions identified in this study have had the potential to contribute to tumor initiation in both patients as it was previously reported (Scott et al. 2016; Miki et al. 1992). But as said,

we do agree that some toning is good here and have done this as requested by the Referee. We think that the results and conclusions are now well balanced. As always, we are of course happy to hear any additional editorial advice.

REVIEWERS' COMMENTS:

Reviewer #2 (Remarks to the Author):

First of all, I would like to thank the authors for the detailed response to each of this reviewer comments. Overall, the manuscript has improved with the new analysis and, most importantly, the balance between results and conclusions seems now more appropriate as some statements have been toned down. This is an important study in a large cohort; however, remaining open questions are: (i) What is the basis for the preferential retrotransposon insertion in fragile sites (e.g., LPR1B) and are these important to initiate colorectal tumorigenesis? (ii) Could the inactivation of APC by retrotransposition be functionally assessed? In fact, L1 insertions have been previously shown in colorectal cancer already. Additionally, insertions in other genes such as FAT4 might as well be of importance, since this gene is also mutated in colorectal cancer. Finally, (iii) does the association between retrotransposon insertion and poor survival holds significant in an independent cohort? These are important questions that will need to be assessed in follow-up studies to further prove the notion that retrotransposon insertions can in fact initiate tumorigenesis and potentially have a prognostic clinical value not only in colorectal, but also in other cancer types.

REVIEWERS' COMMENTS:

Reviewer #2 (Remarks to the Author):

First of all, I would like to thank the authors for the detailed response to each of this reviewer comments. Overall, the manuscript has improved with the new analysis and, most importantly, the balance between results and conclusions seems now more appropriate as some statements have been toned down. This is an important study in a large cohort; however, remaining open questions are:

(i) What is the basis for the preferential retrotransposon insertion in fragile sites (e.g., LPR1B) and are these important to initiate colorectal tumorigenesis?

We thank the reviewer for this valuable suggestion and we have added a sentence in the discussion highlighting that the molecular basis for preferential retrotransposition in fragile sites and whether those are important for the tumorigenic process remain as open questions:

“Yet, the molecular basis of frequent retrotransposition in fragile sites, in particular LRP1B, and whether these insertions are important for the tumorigenic process remain as open questions.”

(ii) Could the inactivation of APC by retrotransposition be functionally assessed? In fact, L1 insertions have been previously shown in colorectal cancer already. Additionally, insertions in other genes such as FAT4 might as well be of importance, since this gene is also mutated in colorectal cancer.

We have also added a sentence in the discussion stating that the inactivation of APC by retrotransposition should be functionally assessed in future studies and that other insertions in known CRC genes should be further investigated:

“... although the inactivation of APC by retrotransposition should be functionally assessed in future studies. In addition, we identified recurrent intronic retrotranspositions in other genes frequently mutated in CRC, such as FAT4, and whether these insertions are important for the tumorigenic process remains to be investigated.”

Finally, (iii) does the association between retrotransposon insertion and poor survival holds significant in an independent cohort?

We have also updated the discussion following the advice of the reviewer:

“Although, further studies need to confirm the prognostic value of retrotransposon insertions not only in CRC, but also in other cancer types.”

These are important questions that will need to be assessed in follow-up studies to further prove the notion that retrotransposon insertions can in fact initiate tumorigenesis and potentially have a prognostic clinical value not only in colorectal, but also in other cancer types.